# Hydrogel-Assisted 3D Volumetric Hotspot for Sensitive Detection by Surface-Enhanced Raman Spectroscopy

**DOI:** 10.3390/ijms23021004

**Published:** 2022-01-17

**Authors:** Soo Hyun Lee, Sunho Kim, Jun-Yeong Yang, ChaeWon Mun, Seunghun Lee, Shin-Hyun Kim, Sung-Gyu Park

**Affiliations:** 1Department of Nano-Bio Convergence, Korea Institute of Materials Science, 797 Changwondae-ro, Changwon 51508, Korea; sh_lee@kims.re.kr (S.H.L.); yjy8184@kims.re.kr (J.-Y.Y.); apple1025@kims.re.kr (C.M.); seunghun@kims.re.kr (S.L.); 2Department of Chemical and Biomolecular Engineering, Korea Advanced Institute of Science and Technology (KAIST), Daejeon 34141, Korea; shkim1020@kaist.ac.kr

**Keywords:** surface-enhanced Raman spectroscopy, volumetric hotspot engineering, maskless plasma etching, hydrogel encapsulation

## Abstract

Effective hotspot engineering with facile and cost-effective fabrication procedures is critical for the practical application of surface-enhanced Raman spectroscopy (SERS). We propose a SERS substrate composed of a metal film over polyimide nanopillars (MFPNs) with three-dimensional (3D) volumetric hotspots for this purpose. The 3D MFPNs were fabricated through a two-step process of maskless plasma etching and hydrogel encapsulation. The probe molecules dispersed in solution were highly concentrated in the 3D hydrogel networks, which provided a further enhancement of the SERS signals. SERS performance parameters such as the SERS enhancement factor, limit-of-detection, and signal reproducibility were investigated with Cyanine5 (Cy5) acid Raman dye solutions and were compared with those of hydrogel-free MFPNs with two-dimensional hotspots. The hydrogel-coated MFPNs enabled the reliable detection of Cy5 acid, even when the Cy5 concentration was as low as 100 pM. We believe that the 3D volumetric hotspots created by introducing a hydrogel layer onto plasmonic nanostructures demonstrate excellent potential for the sensitive and reproducible detection of toxic and hazardous molecules.

## 1. Introduction

Surface-enhanced Raman spectroscopy (SERS) is a powerful and promising technique for identifying molecular fingerprints corresponding to molecules’ vibrational energy states, enabling rapid, contactless, sensitive, label-free, and reliable chemical and biomedical analyses [1,2,3,4,5,6,7]. The amplification of Raman signals originates predominantly from an interaction of incident light with excited electron clouds of noble-metal (e.g., Ag, Au, and Cu) nanostructures (NSs), which is known as the localized surface plasmon resonance (LSPR) effect. Therefore, extensive effort has been devoted to designing elaborate plasmonic hotspots with sub-10 nm scales [8,9,10,11,12,13]. Despite its outstanding sensitivity, the SERS technique has not been widely adopted in practical fields because of complicated fabrication processes, high costs, and poor signal uniformity and reproducibility.

Maskless plasma etching has emerged as an alternative to existing high-precision and complex nanopatterning methods because of its simplicity, cost-effectiveness, low-temperature process, and excellent throughput [14,15,16,17]. For polymer substrates in particular, this technique is favorable for fabricating NSs with high-areal density such as nanotunnels [18], nanodimples [19,20], and nanopillars [17,21] without mask patterning. The formation of such NSs is fundamentally related to complex surface dynamics, including crystallinity-dependent etching, surface migration, agglomeration, and coalescence [22,23]. The geometric properties of NSs are also conveniently tunable through adjustment of the radio-frequency (RF) power, operating pressure, injection gas, and processing time [21,22]. However, the structures of many commercial polymers undergo thermal deformation during SERS measurements, which limits reliable SERS analyses [24,25]. To overcome this problem, researchers have investigated polyimide (PI) as a polymer-based SERS substrate because of its good thermal properties (e.g., glass transition temperature of 300 °C) and chemical/abrasion resistance [26].

Although numerous SERS platforms have been reported, the signals obtained using these platforms are generally collected from molecules adsorbed on two-dimensional (2D) surfaces in hotspots. Positioning molecules in hotspot volume gives a promise towards high sensing performance, but there has been a lack of advanced methodology creating elaborated structures with porosity and sub-nanoscale in narrow active regions. Meanwhile, hydrogels have demonstrated strong potential in molecular capturing and filtering systems because of their large void fraction, controllable mesh size and charge, and hydrophilicity [27,28,29,30]. The dynamic crosslinking under exposure is benefitable for the formation of hydrogel matrix in tiny NSs. Moreover, the high water retention capacity allows the effective transportation and adsorption of molecules dispersed in a solvent, especially water, which is highly compatible with the chemical and biomedical sensing applications. Inspired by the aforementioned factors, the desirable SERS substrates combined with hydrogel networks can provide molecular-concentrating sites for analytes in a three-dimensional (3D) volume among plasmonic NSs; in particular, they provide greater opportunity for molecular adsorption at 3D volumetric hotspots along with NSs with a high aspect ratio (e.g., nanopillars) [13,31]. Accordingly, the sensitivity and limit-of-detection (LOD) are also able to be substantially improved. The infusion of target analytes into complex mixtures through a pore-size-modified hydrogel layer also contributes to selective and multiplex sensing [32,33,34].

In the present work, we prepared a highly sensitive and reproducible 3D volumetric SERS platform with a hydrogel-coated metal film (Au/Ag) over PI nanopillars (MFPNs). The areal density and aspect ratio of the PI nanopillars (PNs) were optimized via a two-step plasma treatment (CF_4_ plasma followed by Ar plasma). The plasmonic hotspots were formed by deposition of a metal layer of appropriate thickness. Subsequent hydrogel encapsulation enables target molecules captured in 3D hydrogel mesh as well as 2D metal surfaces, improving SERS sensitivity. The enhancement factor (EF), LOD, and reproducibility of the hydrogel-coated MFPNs were estimated using Cyanine5 (Cy5) acid dyes and compared with those of hydrogel-free MFPNs. To theoretically verify the optical characteristics of hydrogel-coated MFPNs, the electric (*E*)-field profile was calculated on the basis of the finite-difference time-domain (FDTD) method.

## 2. Results

### 2.1. Structural and Morphological Properties of Hydrogel-Coated MFPNs

The hydrogel-encapsulated MFPNs with the 3D volumetric hotspots were fabricated via two-step maskless plasma etching and hydrogel coating (Figure 1). To design the highly ordered PNs using the maskless plasma etching, we preliminarily investigated the influence of etching gas on the surface morphology and wettability of a PI film (Appendix A). Plasma etching is a combination of physical and chemical etching components (Appendix A). Under CF_4_ plasma etching, CF_4_ dissociates into the fluorine (F) atoms and CF*_x_* radicals (1 ≤ *x* ≤ 3). The F atoms contribute to the breaking of C–C/C–H bonds, the desorption of chemically reacted areas, and the displacement of C–H bonds to C–F bonds, whereas the CF*_x_* radicals participate in the deposition of a fluorinated carbon layer [22,35,36]. Compared with the etching rate in the lateral direction, that in the vertical direction is relatively low because of competition between the desorption and deposition processes. In addition, nonpolar C–F*_x_* bonds tend to inhibit interaction with water molecules. Under Ar plasma etching, Ar atoms/ions collide with the surface atoms and then break their structures, a process referred to as sputtering. The potential-driven Ar atoms/ions (i.e., directionality) attribute to the formation of anisotropic NSs. The polymer atoms with broken bonds at exposed areas represent polarity, resulting in hydrophilicity. Such a phenomenon is consistent with experimental results. The areal density and contact angle of the CF_4_-plasma-etched PNs were approximately 135 ± 10 μm^−2^ and 54.1°, whereas those of the Ar-plasma-etched PNs were 271 ± 16 μm^−2^ and 4.3°, respectively. With the two-step maskless plasma etching (CF_4_ plasma and subsequent Ar plasma), PNs with substantial areal density (30 ± 2 μm^−2^), a hydrophilic surface (4.1°), and a high aspect ratio were obtained and subsequently used as SERS substrates.

Figure 1a,b shows field-emission scanning electron microscopy (FE-SEM) images of the MFPN SERS platform without and with the hydrogel skin. The optimal PNs were obtained over a large area (Figure 1a). Their areal density and aspect ratio were predominantly controlled by the CF_4_ plasma and Ar plasma, respectively. After the metal deposition and hydrogel encapsulation, NSs were retained without substantial deformation or degradation (Figure 1b), which we attributed to the excellent structural adaption of the Ag and hydrogel networks. The crystalline and compositional properties of MFPNs were confirmed by field-effect transmission electron microscopy (FE-TEM) observations (Figure 1c,d). Approximately 1 μm-thick hydrogel completely covered the MFPNs and infiltrated the 3D interstitial space between the MFPNs. We also observed that the thermally deposited Ag layer covered the surface of the PNs conformally and smoothly. The formation of metal NSs is closely related to cohesive energy. In general, the matter is becoming softer when its cohesive energy or surface energy decreases. The cohesive energy and surface energy of Ag were ~2.95 eV and ~1.20 J m^−2^, respectively, whereas those of Au were 3.81 eV and 1.54 J m^−2^, respectively, indicating that the Ag has the better structural adaptive potential [37,38]. The PNs with high surface energy (hydrophilic) were wetted by the Ag layer as soft matter. The high crystallinity of the Ag layer was confirmed by the regular lattice fringes with a *d*-spacing of 2.35 Å, corresponding to the (111) plane of Ag (Figure 1e). The bright dot patterns also indicated the single-crystalline nature (Figure 1f). No indication of Au clusters was observed because (i) the actual Au thickness of ≤2 nm was insufficient for the formation of crystalline structures and (ii) the Au seeds were immersed by the diffused Ag layer. These characteristics were further confirmed by the elemental mapping results (Figure 1g). Both C and O were observed in the hydrogel and the PI. The plasmonic resonance of the MFPNs was confirmed by dark-field optical microscopy. LSPR excitation of plasmonic NSs is known to induce more efficient resonant Rayleigh scattering [39,40,41]. Vivid and different colors were observed, corresponding to wavelength-selective Rayleigh scattering under light illumination. Upon excitation of the plasmonic resonance, strong Rayleigh scattering around MFPNs was generated and could be detected in the far-field (Figure 1h). Because of structural heterogeneities of the MFPNs, broad Rayleigh scattering spectra were recorded across the entire visible wavelength range, whereas no scattering effect was observed over the PN substrate. The induced LSPR effect was also theoretically investigated based on the FDTD method. The geometrical parameters of simulation model were extracted from the scanning TEM (STEM) image (Figure 1g). It was found that the height of MFPNs and Ag layer (bottom) was estimated to be 350 nm and 150 nm, respectively. The MFPNs were inclined with respect to the substrate surface and their interstitial distance (i.e., hotspot) was about ≤5 nm. The 150 nm thick Ag layer did not allow the transmission and confinement of incident light due to the shallow skin depth (2.9 nm at 633 nm). The hotspot was realized by the PNs covered by the Ag with structural randomness. The designed model with simplified parameters demonstrated good agreement with the STEM image. As a result, a strongly confined *E*-field in 3D volumetric hotspots between the MFPNs was observed (Figure 1i). Notably, the interstitial nanogaps among high-aspect-ratio plasmonic nanopillars created a strong *E*-field concentration along the *z*-direction, indicating that the maximum |*E*_loc_|/|*E*_0_| is 52.2. Herein, the |*E*_loc_| is the confined *E*-field intensity in hotspot and |*E*_0_| is the incident wave intensity. Therefore, the molecular concentration at the hotspot regions is critical to enhancing the SERS signal and the corresponding sensitivity of the SERS sensors.

### 2.2. SERS Activities of Hydrogel-Coated MFPNs

We investigated the SERS performance of the hydrogel-coated MFPNs by analyzing the quantitative Raman signals of the Cy5 acid (Figure 2a,b). The dye solutions were prepared over six decades of dilution (10 pM to 10 µM). The spectral intensity was gradually decreased with decreasing dye concentration. The characteristic spectral features of the Cy5 acid were clearly detected even at the low concentration of 100 pM. Among these features, the most intense peak at 1352 cm^−1^, corresponding to typical methine chain vibrations of indocarbocyanines [42,43], was attributed to a representative peak and was used for further analyses. The SERS EF of hydrogel-coated MFPNs was evaluated on the basis of the Raman intensity ratio between the plasmonic and nonplasmonic structures (Figure 2c). The EF was calculated by the following equation [7]:(1)$$\mathrm{EF}=\left(\frac{I_{\mathrm{SERS}}}{I_{\mathrm{bare}}}\right)\left(\frac{N_{\mathrm{bare}}}{N_{\mathrm{SERS}}}\right)$$

Umber of molecules producing the *I*_SERS_ and *I*_bare_ intensities, respectively. Under the assumption of homogeneous distributions of Cy5 acid dyes over the surfaces, the average number of molecules (*N*) can be described using the equation of
(2)$$N=\left(N_{A}\times c\times \frac{V}{A_{\mathrm{sub}}}\right)A_{\mathrm{laser}}$$
where *N*_A_, *c*, *V*, *A*_sub_, and *A*_laser_ are Avogadro’s constant, the dye concentration, the volume of Cy5 acid solution, the surface area covered by the solution, and the area of the laser spot, respectively. Herein, the same experimental conditions were used for both the SERS and bare platforms; thus, the parameters for *N*_A_, *V*, *A*_sub_, and *A*_laser_ could be discarded. The equation for EF could therefore be simplified as
(3)$$\mathrm{EF}=\left(\frac{I_{\mathrm{SERS}}}{I_{\mathrm{bare}}}\right)\left(\frac{c_{\mathrm{bare}}}{c_{\mathrm{SERS}}}\right).$$

For the bare signal, the Raman spectrum for hydrogel-coated PNs dipped in 100 µM Cy5 acid solution was recorded. At 1352 cm^−1^, *I*_SERS_ and *I*_bare_ were estimated to be 52 and 25, respectively, resulting in an experimental EF value of 2.1 × 10^6^ (Figure 2c). Because the EF is approximately proportional to the fourth power of the *E*-field, on the basis of Figure 1i, the theoretical EF was evaluated to be 7.4 × 10^6^. Reasonable EFs were obtained despite the difficulty in matching the geometric and optical parameters between the actual and modeled structures. To validate the hydrogel functionality, we compared these behaviors with those of hydrogel-free MFPNs (Figure 2d). A comparable tendency related to the concentration of the Cy5 acid was observed. The hydrogel-coated MFPNs exhibited excellent sensitivity (EF of 2.1 × 10^6^ and LOD of 100 pM) compared with the hydrogel-free MFPNs (EF of 3.7 × 10^4^ and LOD of 10 nM).

### 2.3. Reproducibility of Hydrogel-Coated MFPNs

Practical assays require homogeneous SERS signals that arise from any random spot on a plasmonic structure. The spectra for Cy5 acid on MFPNs with and without the hydrogel skins were acquired using a Raman mapping image system equipped with an *x*–*y* stage remotely controlled by a computer (Figure 3a). The mapping scan was performed in an area of 25 × 25 μm^2^ in intervals of 2.5 μm (the interval should be larger than laser spot diameter of 1.9 μm). For the 10 µM Cy5 acid on hydrogel-coated MFPNs, the overall spectra (100 points) displayed high uniformity through the entire measurement range (Figure 3b). At the characteristic peak of 1352 cm^−1^, the Raman intensities were distributed within two standard deviations from the average (Figure 3c). On the basis of these intensities, the signals with a relative standard deviation (RSD) of 10.0% exhibited the excellent reproducibility of hydrogel-coated MFPNs. This was also confirmed in the reconstructed Raman mapping image in Figure 3d. For comparison, Raman mapping was also performed for 10 µM Cy5 acid on hydrogel-free MFPNs (Figure 3e). The Raman mapping for the hydrogel-free MFPNs indicated uniform signals with an RSD of 9.0%. This strong signal stability of MFPNs was attributed to the reasonably high periodicity of the PNs prepared by maskless plasma etching.

### 2.4. Optimization of Dipping Time for Efficient SERS Analysis

The SERS signals are proportional to the probe molecules adsorbed in the hotspots. For the MFPNs with and without the hydrogel layer, we optimized the dipping time to achieve high SERS performance in a relatively short time. The SERS intensities of the MFPNs with and without the hydrogel were measured at 1352 cm^−1^ after the MFPNs were immersed in 10 µM Cy5 acid solution for various times (*t* = 15, 30, 45, 60 min) (Figure 4). For both the MFPNs with and without the hydrogel matrix, with increasing dipping time, the peak intensity increased, whereas the slope efficiency (*I*_Raman_/*t*) was gradually degraded. The intensity almost reached saturation when the dipping time was 60 min. Therefore, the optimum dipping time was set to be 60 min. We also observed that the Raman intensity of the hydrogel-coated MFPNs was consistently higher than that of the hydrogel-free MFPNs because the hydrophilic hydrogel networks allowed the effective diffusion of dyes into the 3D volumetric hotspots.

## 3. Discussion

The development of hotspots with high coherency and density is a key factor governing the SERS activity. The maskless plasma etching is considered a convenient fabrication method for polymer NSs with excellent tunability in geometry (i.e., shape, areal density, and aspect ratio) and wettability. Since a hotspot is defined as the dielectric spaces between the plasmonic NSs in proximity (typically sub-10 nm), the areal density of NSs is directly associated with the hotspot density as well as the SERS sensitivity. Despite structural randomness, a remarkable hotspot population also provides an opportunity to increase the order of regularity in NS arrays [7,13,19,44]. Consequently, it is available to achieve SERS signals with high uniformity, leading to sensitive, reproducible, repeatable, and reliable molecular detection. In the present study, the highly populated MFPNs (30 ± 2 μm^−2^) obtained by our novel two-step maskless plasma treatment method demonstrated excellent SERS spectral stability, with an RSD of 9.0% (Figure 3e).

The aspect ratio of the NSs and hydrogel attributed to the formation of 3D volumetric hotspots. Although the pillar-like structures produce strongly confined *E*-fields along their longitudinal axis, the modified surface tension across the narrow regions inhibits effective molecular penetration. To characterize the molecular transport in addition to the volumetric distributions, we used the hydrogel matrix. In the present study, the MFPNs with a high aspect ratio (Figure 1g and Appendix A) were conformally wetted by the Ag layer (Figure 1b–d,g), which acted as soft matter because of its intrinsic material properties. The plasmonic activities of MFPNs were experimentally (Figure 1h) and theoretically (Figure 1i) demonstrated. The hydrogel solution was diluted in ethanol for better infiltration, resulting in MFPNs fully covered with hydrogel networks (Figure 1b–d). The solute diffusion through hydrogel is described with the following three traditional theories: hydrodynamic theory, free volume theory, and obstruction theory. The mesh size of hydrogel matrix was larger than 4.6 nm according to our previous study [32]. Based on the reported hydrodynamic radius of Cy5 (0.81 nm) [45], the diffusivity of Cy5 acid was estimated to be 269 μm^2^/s, which is very high enough to diffuse into the whole hydrogel thickness. Therefore, it is postulated that the Cy5 acid dyes dissolved in water were diffused into the MFPNs through the hydrogel matrix (i.e., hydrodynamic theory). This full-coverage enabled the target molecules to locate not only on the substrate surface but also in the spaces. Therefore, the introduction of a hydrogel layer into the plasmonic NSs is important for enhancing the SERS performance (i.e., the sensitivity and reproducibility). The hydrogel-encapsulated MFPNs exhibited an EF of 2.1 × 10^6^, LOD of 100 pM, and RSD of 10.0%, as compared with those of hydrogel-free MFPNs (EF of 3.7 × 10^4^, LOD of 10 nM, and RSD of 9.0%) (Figure 2 and Figure 3).

In conclusion, the hydrogel-assisted MFPNs with 3D volumetric hotspots provide great insight into the development of a highly sensitive, reproducible, and reliable SERS platform for the detection and identification of small analytes.

## 4. Materials and Methods

### 4.1. Fabrication Procedures of Hydrogel-Coated MFPNs

A 50 μm-thick PI film with dimensions of 3 × 3 cm^2^ (Isoflex Kespi, Siheung, Korea) was used as a SERS substrate after its protective film was removed. To obtain the desired PNs, a two-step maskless plasma etching technique was conducted by reactive-ion etching using 13.56 MHz RF plasma equipment (LAT, Osan, Korea). At the initial stage, the base pressure of the chamber was set to be 7.0 mTorr. The CF_4_ plasma treatment was applied with a gas flow rate of 3 sccm, working pressure of 50 mTorr, and RF power of 100 W for 120 s. After the recovery of the base pressure, the Ar plasma etching with a gas flow rate of 3 sccm, working pressure of 30 mTorr, and RF power of 100 W was carried out for 120 s. To produce the plasmonic hotspots, a metal film was deposited over the PNs, resulting in MFPNs. A 200 nm-thick Ag layer was deposited at a rate of 1.8 Å s^−1^ using a thermal evaporator (LAT, Osan, Korea). A 5 nm-thick Au layer was subsequently deposited at a rate of 2.0 Å s^−1^ under an applied RF power of 100 W using a sputtering system (LAT, Osan, Korea). The as-fabricated MFPNs were diced into individual SERS platforms with dimensions of 5 × 5 mm^2^.

A hydrogel solution was prepared by dissolving poly(ethylene glycol) diacrylate (PEGDA; MW 700 g mol^−1^, Sigma-Aldrich) and 1-[4-(2-hydroxyethoxy)-phenyl]-2-hydroxy-2-methyl-1-propane-1-one (Irgacure 2959, BASF) as a photoinitiator in deionized (DI) water. To control the surface area of the hydrogel skin, the solution was highly diluted with ethanol. To render the hydrogel layer, a 1 μL droplet (0.7 *w/w*% PEGDA, 0.6 *w/w*% DI water, and 0.005 *w/w*% photoinhibitor) was placed on the surface of MFPNs and evaporated for 5 min in air. Under UV exposure, the PEGDA molecules were crosslinked (i.e., via free-radical polymerization). The hydrogel-coated MFPNs were rinsed with DI water to remove the excess monomer residues and then dried in air.

### 4.2. Characterization

The morphological properties of hydrogel-coated MFPNs were observed by FE-SEM (JSM−7900F, Jeol, Tokyo, Japan). The crystallinity and elemental composition were explored via focused ion beam (Quanta 3D FEG, FEI, Eindhoven, Netherlands) milling at 0.1 nA and 30 kV and by FE-TEM (JEM−2100F, Jeol, Tokyo, Japan) in conjunction with energy-dispersive X-ray analysis (EDX) at an accelerating voltage of 200 kV. The SERS activities of the hydrogel-coated MFPNs were evaluated using a Raman microscope (HEDA, NOST, Sungnam, Korea) equipped with a 633 nm laser diode (0.16 mW) and a 20× objective lens (NA = 0.45). For the quantitative studies, the samples were immersed into Raman probe solutions (10 pM to 10 μM), followed by ambient drying. Their spectra in the wavenumber range 400–1800 cm^−1^ were acquired for 1 s at five random points. Cy5 acid (C_32_H_39_ClN_2_O_2_; 98%, Broadpharm) was used as a Raman probe molecule.

### 4.3. Theoretical Computation

The *E*-field profiles across the MFPNs were theoretically calculated using the FDTD software (version 2021 R1.2, Ansys Lumerical, Seoul, Korea). The geometric parameters were extracted from the STEM image. The Ag nanopillar arrays covering PNs were depicted with surface roughness. The hydrogel skin was used as the top domain, whereas the Ag film was used as the bottom domain. For the boundary condition, the perfectly matched layer was applied to all planes to eliminate the reflection at an interface. The *E_x_*-polarized plane wave at *λ* = 633 nm (*E*_0_) was produced from the active port and was incident to the MFPNs in the normal direction. The intensity of *E*_0_ was determined to be 1 V m^−1^. The mesh size was fixed at 0.5 nm. For computations, the dielectric constant of the Ag and Au was taken as *ε*_Ag_ = −18.8626 + 2.3267i and *ε*_Au_ = −10.8911 + 0.7948, respectively, and the refractive index of the hydrogel was set to 1.47.

## Data Availability

Not applicable.

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
