# Peer review of "Hydrogel-Assisted 3D Volumetric Hotspot for Sensitive Detection by Surface-Enhanced Raman Spectroscopy"

_ijms, 2022, doi:10.3390/ijms23021004_

Round 1

Reviewer 1 Report

The manuscript of the paper entitled « Hydrogel-assisted 3D volumetric hotspot for sensitive detection by surface-enhanced Raman spectroscopy» presents a multi-steps methodology to produce SERS substrate.

In my opinion this manuscript can be published in the journal International Journal of moecular Science after major revisions.

The fabricated nanostructures are very nice and the SERS results are very interesting but some issues should be properly adressed and improved before publication.

1 - The size and the shape of the nanostructures should be discussed :

On the TEM and on the SEM images (Figure 1) the lentgh of the nanopillars covered with Ag and Au can be estimated at 350-400 nm. The nanopillars are not perpendicular to the surface and seem to be entangled. The calculation pictures show straight and shorter (200 nm) nanopillar with constant gap between them.

2 - Plasmonic response

The authors write « Practical assays require homogeneous SERS signals that arise from any random 210 spot on a plasmonic structure ».  As the nanostructures are not the same on the whole surface of the substrate are the plasmonic properties homogeneous (Figure 1c) ?  

3 - SERS and contribution of hydrogel

The authors write « Because hotspots are defined as the dielectric spaces between the plasmonic NSs in proximity ». This definition is not sufficient and should be developped. On this point the contribution of the hydrogel is not clear in Figure 3a and in the section discussion and shloud also be developped.

The distance between the top of the hydrogel and the top of the nanopillars is very important compared to the « thickness » of the nanostructures (Figure 1e). Do the authors have a model to explain how the probe molecule interact with the hydrogel ?

For the SERS measurements the way to calcul the EF with the Volume of the droplet should be improved beacause of the interaction of the droplet with the hydrogel.

Author Response

<General Comment> The manuscript of the paper entitled “Hydrogel-assisted 3D volumetric hotspot for sensitive detection by surface-enhanced Raman spectroscopy” presents a multi-steps methodology to produce SERS substrate.

In my opinion this manuscript can be published in the journal International Journal of Molecular Science after major revisions.

The fabricated nanostructures are very nice and the SERS results are very interesting but some issues should be properly addressed and improved before publication.

<Response> We thank the referee for giving us these valuable comments. We have addressed the concerns during the revision. Please check the revised manuscript.

<Comment 1> The size and the shape of the nanostructures should be discussed :

On the TEM and on the SEM images (Figure 1) the length of the nanopillars covered with Ag and Au can be estimated at 350-400 nm. The nanopillars are not perpendicular to the surface and seem to be entangled. The calculation pictures show straight and shorter (200 nm) nanopillar with constant gap between them.

<Response> We really appreciate your valuable comment. The geometrical parameters used in simulation were extracted from the TEM analysis. It was noticed that i) the average height of MFPNs was estimated to be approximately 350 nm, ii) the bottom area (height ≤ 150 nm) was filled by the Ag layer, and iii) nanopillars were inclined respect to the substrate surface.

In this study, the theoretical calculation of E-field distribution was carried out to prove the induced LSPR effect at the interstitials between MFPNs. The E-field strength and distribution of hotspots play a key role to determine the SERS performance. The hotspot scale of MFPNs platform was estimated to be ≤ 5 nm. For calculations, the MFPNs structures were described to the have simplified 3D model; the straight PI nanopillars covered by the Ag nanopillars with the structural randomness, which demonstrated good agreement with the TEM image (Figure 1i). The bottom Ag layer with thickness of 150 nm did not allow the transmission and/or confinement of incident light due to its shallow skin depth (approximately 2.9 nm at 633 nm). Therefore, the E-field profile of MFPNs in the height range of ≥ 250 nm was represented (Figure 1d).

<Comment 2> Plasmonic response

The authors write « Practical assays require homogeneous SERS signals that arise from any random spot on a plasmonic structure ». As the nanostructures are not the same on the whole surface of the substrate are the plasmonic properties homogeneous (Figure 1c) ?

<Response> Since the SERS signal is mainly collected from molecules adsorbed in the hotspot region, its intensity is proportional to a number of hotspots (i.e. areal density of plasmonic nanostructures) in an illumination area exposed by laser. With this regard, the two-step maskless plasma etching technique was employed to produce the PNs with high areal density (30 ± 2 μm-2) (Figure 1a-b and Figure S1c) which contributed to the high order of periodicity. The Raman mapping (25 × 25 μm2 with intervals of 2.5 μm) of MFPNs was performed to experimentally verify the signal uniformity (Figure 3b-e). From the SERS spectra (100 points), the sufficient signal reproducibility was confirmed with the RSD value of ≤ 10 %, demonstrating its applicability in practical fields.

<Comment 3> SERS and contribution of hydrogel

The authors write « Because hotspots are defined as the dielectric spaces between the plasmonic NSs in proximity ». This definition is not sufficient and should be developed. On this point the contribution of the hydrogel is not clear in Figure 3a and in the section discussion and should also be developed.

The distance between the top of the hydrogel and the top of the nanopillars is very important compared to the « thickness » of the nanostructures (Figure 1e). Do the authors have a model to explain how the probe molecule interact with the hydrogel?

For the SERS measurements the way to calculate the EF with the Volume of the droplet should be improved because of the interaction of the droplet with the hydrogel.

<Response> It is well-known that, in the SERS, the induced LSPR effect can be observed in the dielectric spaces (called as hotspot) between plasmonic NSs and typically, the hotspot scale is in the range of 1-10 nm. On this basis, the same field confinement is induced from the MFPNs with and without hydrogel. The difference in 2D and 3D hotspot is determined by the sites where the analyte can be adsorbed (Figure 3a). In the case of MFPNs without hydrogel, the presence of molecules is allowed only on the surface of MFPNs. With the introduction of hydrogel into MFPNs, the molecules are adsorbed not only on the surface of MFPNs but also to the hydrogel matrix.

There are three mechanism for molecular diffusion through hydrogel matrix; i) hydrodynamic theory, ii) free volume theory, and iii) obstruction theory. Herein, the rs, rFV, and ξ indicates the hydrodynamic radius of solute in a pure liquid, free volume void radius at atomic scale, and size of hydrogel mesh, respectively. In the case of rFV < rs < ξ, the transportation of solute into hydrogel layer is performed by liquid molecules. With the atomic scale rs, the solute travels through the spaces between molecules. When the rs ≥ ξ, the solute can not penetrate into the hydrogel layer. In this study, the ξ becomes larger than 4.6 nm according to our previous study. Based on the Stokes-Einstein-Debye equation under the assumption of a sphere shape of dye, the rs of Cy5 is reported to be 0.81 nm [a] and consequently, the diffusivity of Cy5 acid is expected to be ~ 269 μm2/s. Therefore, we postulate that the Cy5 acid dyes dissolved in water are diffused into the MFPNs through the hydrogel layer. We added the diffusion behaviors in the revised manuscript.

Page 8, The solute diffusion through hydrogel is described with three traditional theories; hydrodynamic theory, free volume theory, and obstruction theory. The mesh size of hydrogel matrix was larger than 4.6 nm according to our previous study. Based on the reported hydrodynamic radius of Cy5 (0.81 nm)[45], the diffusivity of Cy5 acid was estimated to be 269 μm2/s, which is very high enough to diffuse into the whole hydrogel thickness. Therefore, it is postulated that the Cy5 acid dyes dissolved in water were diffused into the MFPNs through the hydrogel matrix (i.e., hydrodynamic theory).

The corresponding reference is added as below:

[45] Gebhardt, C.; Lehmann, M.; Reif, M.M.; Zacharias, M.; Gemmecker, G.; Cordes, T. Molecular and Spectroscopic Characterization of Green and Red Cyanine Fluorophores from the Alexa Fluor and AF Series. ChemPhysChem 2021, 22, 1566–1583. https://doi.org/10.1002/cphc.202000935.

The SERS MFPNs platform with and without hydrogel layer were immersed into the Cy5 acid dissolved DI water in this study. To avoid the confusion, the V is used instead of the Vdroplet. The change in the revised manuscript is as shown in below:

Page 5, Eq(2),

Page 5, where NA, c, V, Asub, and Alaser are Avogadro’s constant, the dye concentration, the volume of Cy5 acid solution, the surface area covered by the solution, and the area of the laser spot, respectively.

Page 6, the parameters for NA, V, Asub, and Alaser could be discarded.

Reviewer 2 Report

The issue of the present manuscript is very interesting and of much value. The paper is written very well.

However, I recommend taking care of the figures` localization/distribution along with the paper (to be able to read the paper and check the results in the figures more fluently/simultaneously/easily). I had also problems following the text of the manuscript because of the inserted figures.

Further, for example, the plots in Figure 2c could/should be increased (there is unnecessary free space and the plots are too small).

I also recommend mentioning/clarifying what kind of vibration is assigned to the characteristic wavenumber (the most intense peak) discussed in the paper, observed at 1352 cm-1.

Author Response

<General Comment> The issue of the present manuscript is very interesting and of much value. The paper is written very well.

<Response> We thank the referee for carefully reading our manuscript to help us improve it. We really appreciate your positive feedback.

<Comment 1> I recommend taking care of the figures` localization/distribution along with the paper (to be able to read the paper and check the results in the figures more fluently/simultaneously/easily). I had also problems following the text of the manuscript because of the inserted figures.

<Response> As the reviewer’s comment, the position of figures is rearranged in the revised manuscript.

<Comment 2> Further, for example, the plots in Figure 2c could/should be increased (there is unnecessary free space and the plots are too small).

<Response> As the reviewer’s comment, the SERS spectra in Figure 2c were increased in the revised manuscript as shown in below:

<Comment 3> I also recommend mentioning/clarifying what kind of vibration is assigned to the characteristic wavenumber (the most intense peak) discussed in the paper, observed at 1352 cm-1.

<Response> The Raman band assignments of carbocyanines have been extensively investigated. For the Cy5 acid, the peak at 1352 cm-1 corresponds to methine chain vibrations. As the reviewer’s comment, the peak assignment is added in the revised manuscript as shown in below:

Page 5: Among these features, the most intense peak at 1352 cm−1, corresponding to typical methine chain vibrations of indocarbocyanines [42,43], was attributed to a representative peak and was used for further analyses.

The corresponding references are added as below:

[42] Sato, H.; Kawasaki, M.; Kasatani K.; Katsumata, M.-a. Raman Spectra of Some Indo-, Thia- and Selena-carbocyanine Dyes. J. Raman Spectrosc. 1988, 19, 129–132. https://doi.org/10.1002/jrs.1250190210.

[43] Novara, C.; Chiadò, A.; Paccotti, N.; Catuogno, S.; Esposito, C.L.; Condorelli, G.; Franciscis, V. De; Geobaldo, F.; Rivolo, P.; Giorgis F. SERS-active metal-dielectric nanostructures integrated in microfluidic devices for label-free quantitative detection of miRNA. Faraday Discuss. 2017, 205, 271. https://doi.org/10.1039/C7FD00140A.

Round 2

Reviewer 1 Report

I Thank the authors for all their answers.

It would have been beneficial to add some answers in the manuscript especially concerning the size of the nanostructures.

Author Response

We thank the referee for giving us the opportunity to improve our manuscript. We have addressed the concern during the revision. Please check the revised manuscript.

As the reviewer’s comment, we added the size of nanostructures in TEM analysis and simulation in the revised manuscript. Since the simulation parameters were extracted from the STEM image, for better legibility, the arrangements in Figure 1 and consequent paragraphs were changed.

Page 3–5, the paragraphs for Figure 1 were rearranged.

Page 4, Figure 1 and its caption were changed as below.

Figure 1. Morphological and crystalline analyses of the hydrogel coated MFPNs. (a−b) FE-SEM images of the MFPNs without and with hydrogel encapsulation. Structural and crystalline properties of the hydrogel-coated MFPNs observed by (c–d) FE-TEM images, (e) a HR-TEM image, and (f) the fast Fourier transform of the image in (e). (g) Compositional analysis with elemental mapping of Ag, Au, C, and O in the hydrogel-coated MFPNs. (h) Scattering intensity of the nonplasmonic PNs and plasmonic MFPNs with and without the hydrogel skin. The photograph of (h) demonstrates the true color of scattered light. (i) Numerically calculated E-field profiles of the hydrogel-coated MFPNs.

Page 4–5, the geometrical information in STEM analysis and simulation was added as below.

The induced LSPR effect was also theoretically investigated based on the FDTD method. The geometrical parameters of simulation model were extracted from the scanning TEM (STEM) image (Figure 1g). It was found that the height of MFPNs and Ag layer (bottom) was estimated to be 350 nm and 150 nm, respectively. The MFPNs were inclined respect to the substrate surface and their interstitial distance (i.e., hotspot) was about ≤ 5 nm. The 150 nm thick Ag layer did not allow the transmission and confinement of incident light due to the shallow skin depth (2.9 nm at 633 nm). The hotspot was realized by the PNs covered by the Ag with structural random-ness. The designed model with simplified parameters demonstrated good agreement with the STEM image. As a result, a strongly confined E-field in 3D volumetric hotspots be-tween the MFPNs was observed (Figure 1i).
